# Molecular Characterization and Antimicrobial Susceptibilities of *Corynebacterium pseudotuberculosis* Isolated from Caseous Lymphadenitis of Smallholder Sheep and Goats

**DOI:** 10.3390/ani13142337

**Published:** 2023-07-18

**Authors:** Hend M. El Damaty, Azza S. El-Demerdash, Norhan K. Abd El-Aziz, Sarah G. Yousef, Ahmed A. Hefny, Etab M. Abo Remela, Asmaa Shaker, Ibrahim Elsohaby

**Affiliations:** 1Department of Animal Medicine, Infectious Diseases, Faculty of Veterinary Medicine, Zagazig University, Zagazig 44511, Egypt; drsara10514@gmail.com (S.G.Y.); ielsohab@cityu.edu.hk (I.E.); 2Agriculture Research Center (ARC), Animal Health Research Institute (AHRI), Zagazig 44516, Egypt; dr.azzasalah@yahoo.com; 3Department of Microbiology, Faculty of Veterinary Medicine, Zagazig University, Zagazig 44511, Egypt; 4Veterinary Hospital, Faculty of Veterinary Medicine, Zagazig University, Zagazig 44511, Egypt; ahmed_vet8_2007@yahoo.com; 5Department of Bacteriology, Mycology and Immunology, Faculty of Veterinary Medicine, Kafrelsheikh University, Kafrelsheikh 33516, Egypt; etab_mh@yahoo.com; 6Department of Biology, College of Science, Taibah University, Medina 42353, Saudi Arabia; 7Department of Microbiology, Veterinary Hospital, Faculty of Veterinary Medicine, University of Sadat City, Sadat City 32897, Egypt; asmaa.shaker@vet.usc.edu.eg; 8Department of Infectious Diseases and Public Health, Jockey Club of Veterinary Medicine and Life Sciences, City University of Hong Kong, Hong Kong SAR 999077, China; 9Centre for Applied One Health Research and Policy Advice (OHRP), City University of Hong Kong, Hong Kong SAR 999077, China

**Keywords:** *C. pseudotuberculosis*, multidrug-resistance, sheep, goats, smallholders, ERIC-PCR, phylogenetic analysis

## Abstract

**Simple Summary:**

The present study aimed to isolate *Corynebacterium pseudotuberculosis* from caseous lymphadenitis (CLA) in smallholder sheep and goats. Thereafter, antimicrobial resistance determinants, virulence characteristics, and phylogenetic analysis of the isolates were investigated. Our results revealed that 24.54% (54/220) of examined animals showed CLA-compatible lesions from which *C. pseudotuberculosis* was isolated. Antimicrobial susceptibility testing revealed that all isolates were resistant to bacitracin and florfenicol. Additionally, 92.6% of the isolates displayed resistance to penicillin and erythromycin, while none of the isolates were resistant to norfloxacin. Interestingly, 16.7% of *C. pseudotuberculosis* isolates recovered from sheep showed vancomycin resistance. Molecular characterization revealed that *PLD*, *PIP*, and *FagA* virulence genes were detected in all examined *C. pseudotuberculosis* isolates. The *bla* (*β*-lactam) resistance gene was present in all isolates. Additionally, 83% of sheep isolates carried the aminoglycoside (*aph(3″)-lb*), chloramphenicol (*cat1*), and bacitracin (*bcrA*) resistance genes. Notably, the glycopeptide (*vanA*) resistance gene was detected in 8% of the sheep isolates as a first report. Enterobacterial Repetitive Intergenic Consensus Polymerase Chain Reaction (ERIC-PCR) genotyping of 10 multi-drug resistant *C. pseudotuberculosis* isolates showed a high similarity index of 83.6% between isolates from sheep and goats. Nucleotide sequence analysis of partial 16S rRNA sequences of these isolates showed 98.83% similarities with biovar *Ovis* of globally available reference sequences on the Genbank database. In conclusion, this study emphasized the high isolation rate of *C. pseudotuberculosis* and its molecular characterization in smallholder sheep and goats in four districts in Egypt.

**Abstract:**

Caseous lymphadenitis (CLA) is a bacterial infection caused by *Corynebacterium pseudotuberculosis* (*C. pseudotuberculosis*) that affects sheep and goats, leading to abscess formation in their lymph nodes. The present study aimed to isolate and identify *C. pseudotuberculosis* from CLA in smallholder sheep and goats, and determine the resistance patterns, virulence, and resistance genes of the isolates. Additionally, genotypic and phylogenetic analysis of the isolates was conducted using ERIC-PCR and DNA sequencing techniques. A cross-sectional study examined 220 animals (130 sheep and 90 goats) from 39 smallholder flocks for clinical signs of CLA. Fifty-four (24.54%) animals showed CLA-compatible lesions, confirmed by *C. pseudotuberculosis* isolation and PCR identification. Sheep had a lower infection rate of CLA (18.46%) compared with goats (33.3%). Antimicrobial susceptibility testing of 54 *C. pseudotuberculosis* isolates to 24 antimicrobial drugs revealed that they were 100% resistant to bacitracin and florfenicol, while none of the isolates were resistant to norfloxacin. A high resistance rate was observed for penicillin and erythromycin (92.6% each). Interestingly, 16.7% of *C. pseudotuberculosis* isolates recovered from sheep showed vancomycin resistance. Molecular characterization of *C. pseudotuberculosis* isolates revealed that *PLD*, *PIP*, and *FagA* virulence genes were present in all examined isolates. However, the *FagB*, *FagC*, and *FagD* genes were detected in 24 (100%), 20 (83%), and 18 (75%) of the sheep isolates, and 26 (87%), 26 (87%), and 18 (60%) of the goat isolates, respectively. The *β*-lactam resistance gene was present in all isolates. Furthermore, 83% of the sheep isolates carried the aminoglycoside (*aph(3″)-lb*), chloramphenicol (*cat1*), and bacitracin (*bcrA*) resistance genes. Among the isolates recovered from goats, 73% were found to contain macrolides (*ermX*), sulfonamide (*sul1*), and bacitracin (*bcrA*) resistance genes. It is worrisome that the glycopeptide (*vanA*) resistance gene was detected in 8% of the sheep isolates as a first report. ERIC-PCR genotyping of 10 multi-drug-resistant *C. pseudotuberculosis* isolates showed a high similarity index of 83.6% between isolates from sheep and goats. Nucleotide sequence analysis of partial 16S rRNA sequences of *C. pseudotuberculosis* revealed 98.83% similarity with biovar *Ovis* of globally available reference sequences on the Genbank database. Overall, our findings might indicate that *C. pseudotuberculosis* infection in smallholders in Egypt might be underestimated despite the significant financial impact on animal husbandry and potential health hazards it poses. Moreover, this study highlights the importance of implementing a sustainable control strategy and increasing knowledge and awareness among smallholder breeders to mitigate the economic impact of CLA.

## 1. Introduction

Caseous lymphadenitis (CLA) is a widespread chronic wasting disease of sheep and goats caused by facultative intracellular *Corynebacterium pseudotuberculosis*. CLA accrues significant financial losses due to reduced wool, meat, and milk production, loss of fertility, culling of affected animals, and their condemnation at the time of slaughter and inspection [1,2]. *C. pseudotuberculosis* is a Gram-positive, non-motile pleomorphic rod that often presents as characteristic Chinese letters in the stained smear [3]. Two biotypes of *C. pseudotuberculosis* exist, the first nitrate-negative strains classified as biotype *Ovis* and the second nitrate-positive strains classified as biotype *Equi*, depending on nitrate-reducing potential [4].

CLA has different forms depending on the location of the abscesses [5]. The most common is the classical “external” form, which is characterized by the formation of abscesses in the superficial lymph nodes. Visceral CLA abscessation, on the other hand, occurs in various lymph nodes and other internal organs [5]. The most commonly infected superficial lymph glands are the parotid, submandibular, prescapular, prefemoral, popliteal, and supramammary lymph glands [6]. *C. pseudotuberculosis* can rarely infect humans. Reported cases are primarily detected in farm workers and veterinarians who have direct contact with sick animals [7].

Bacterial culture is the primary laboratory test used for the diagnosis of CLA, while biochemical and nucleic acid-based detection methods are used for confirming the diagnosis [8]. Serological tests such as ELISA have also been used to identify CLA-infected animals [9]. Molecular typing strategies, including restriction fragment length polymorphisms [10], pulsed-field gel electrophoresis [11], and ribotyping [12], classify *C. pseudotuberculosis* into two biovars but not for further characterization, particularly among biovar *Ovis* isolates, which may be noticeably homogeneous. Thus, for investigating the epidemiological relationships and sources of infection in *C. pseudotuberculosis* isolates from sheep and goats, Enterobacterial Repetitive Intergenic Consensus (ERIC-PCR) has shown good discriminatory power, typeability, and promising results, which consequently improve CLA control trials [13,14,15].

Several virulence genes contribute to the pathogenicity of *C. pseudotuberculosis,* including the phospholipase D (*PLD*) gene, integral membrane protein (*FagA*), iron enterobactin transporter (*FagB*), ATP-binding cytoplasmic membrane protein (*FagC*), and iron siderophore binding protein (*FagD*) [16]. The *PLD* gene encodes the phospholipase D-PLD exotoxin, which is the primary virulence factor in *C. pseudotuberculosis* [17]. This exotoxin is an enzyme that dissociates sphingomyelin, increases vascular permeability, and enables the survival of *C. pseudotuberculosis* in the cells, facilitating the invasion of the body and transport by phagocytes to regional lymph nodes [18]. In addition to the *PLD* gene, downstream putative genes, including *FagA*, *FagB*, *FagC*, and *FagD,* are related to iron regulation and uptake by *C. pseudotuberculosis* [16,19].

The emergence and spread of antimicrobial resistance is a relevant problem, generating inefficient treatment against CLA, its dissemination among small ruminants around the world, and significant economic losses [20]. Previous studies have reported high resistance of *C. pseudotuberculosis* to penicillins, which are the primary drugs for treating CLA in sheep and goats [21,22].

In Egypt, CLA is a significant economic and welfare concern for sheep and goat farmers, as it reduces productivity in infected animals. Additionally, abscesses in the carcass decrease the value of the meat. Studies have reported the prevalence of CLA in sheep and goats from the Nile Delta region of Egypt to range from 3.7% to 32.7% in sheep and 5.6% in goats [23,24]. There have been limited studies investigating the occurrence, distribution, antimicrobial resistance, and genetic characteristics of *C. pseudotuberculosis* infection in smallholder sheep and goats. Therefore, the objectives of the present study were to (i) isolate and identify *C. pseudotuberculosis* from CLA in smallholder sheep and goats; (ii) determine the antimicrobial resistance patterns, virulence, and resistance genes of *C. pseudotuberculosis* isolates; and (iii) conduct genotyping and phylogenetic analysis of the *C. pseudotuberculosis* isolates using ERIC-PCR and DNA sequencing techniques.

## 2. Materials and Methods

### 2.1. Study Design and Population

A cross-sectional study was conducted between February 2021 and January 2022 to investigate the *C. pseudotuberculosis* infection in sheep and goats in smallholder flocks in four districts of Sharkia Governorate, Egypt. Smallholders in the four districts were invited to participate in the study. However, only 39 smallholder flocks agreed to participate. All animals (*n* = 220, comprising 130 sheep and 90 goats) in these flocks were examined for clinical signs of CLA, particularly the presence of enlargement in the superficial lymph nodes (Figure 1).

Purulent material was collected using sterile syringes after the animals were prepared aseptically (by clipping, shaving, and disinfecting the skin). The abscesses were then surgically treated by lancing, antiseptic lavage, and drainage application until complete healing. A systemic course of antibiotics and anti-inflammatory drugs was administered for 5 and 3 successive days, respectively. Duplicate purulent material samples were collected for bacterial culture and direct-sample PCR. Samples were labeled with animal species, ID, date of collection, and districts before being transported to the laboratory on ice for further analysis. Information regarding age, gender, feeding, and skin lesions was collected.

### 2.2. C. pseudotuberculosis Isolation and Identification

Collected purulent material samples were bacteriologically examined following the procedures of Quinn et al. [25]. Briefly, each sample was transferred to 10 mL of brain heart infusion broth (BHI; Oxoid, Hampshire, UK), and incubated at 37 °C for 18 h to stimulate the growth of the microorganisms. A loopful of the pre-enriched sample was then plated onto a blood agar base (Oxoid, Hampshire, UK) enriched with 5% sterile defibrinated sheep blood and incubated aerobically at 37 °C for 48 h. The plates were examined for typical growth, morphological features, and hemolytic characteristics. Gram-positive bacilli in small, white, dry, and crumbly colonies were selected for further identification. Biochemical tests including catalase, urease, nitrate reduction, esculin hydrolysis, and fermentation of glucose, lactose, maltose, sucrose, arabinose, fructose, mannose, trehalose, xylose, and salicin sugars were performed [25,26]. In addition, the suspected colonies were subjected to synergistic hemolysis with *Rhodococcus equi* and antagonistic hemolysis with *Staphylococcus aureus* (ATCC 25923), as described by Cowan and Steel [27].

### 2.3. DNA Extraction and PCR Amplification

The QIAamp DNA Mini kit (Qiagen, Germany: Catalogue no. 51304) was used to extract DNA from *C. pseudotuberculosis* isolates, with modifications to the manufacturer’s recommendations. Briefly, 200 µL of the bacterial suspension were incubated with 10 µL of proteinase K and 200 µL of lysis buffer at 56 °C for 10 min. After incubation, 200 µL of ethanol (100%) were added to the lysate, which was then washed and centrifuged according to the manufacturer’s recommendations. The nucleic acid was eluted with 100 µL of elution buffer provided in the kit. PCR was performed using genus-and species-specific primers (Table 1) to confirm *C. pseudotuberculosis*.

### 2.4. Antimicrobial Susceptibility Testing

*C. pseudotuberculosis* isolates were tested for antimicrobial susceptibilities against 24 commonly used antimicrobial agents using the disc diffusion method on Mueller-Hinton agar with 5% sheep blood (Oxoid, Hampshire, UK). The 24 antimicrobials tested included: penicillin G (PEN, 10 U), novobiocin (NB, 30 μg), amoxicillin/clavulanic acid (AMC, 20/10 μg), ampicillin/sulbactam (SAM, 20 μg), oxacillin (OXA, 5 μg),vancomycin (VAN, 30 μg), piperacillin (PIP, 30 μg), ceftriaxone (CRO, 30 μg), cefuroxime sodium (CXM, 30 μg), cephradine (CEP, 30 μg), bacitracin (BAC, 10 U), ampicillin (AMP, 10 μg), doxycycline (DOX, 5 μg), clindamycin (CLI, 2 μg), nalidixic acid (NA, 30 μg), norfloxacin (NOR, 10 μg), streptomycin (STR, 10 μg), amikacin (AMK, 30 μg), kanamycin (KAN, 30 μg), neomycin (NEO 30 μg), florfenicol (FFC, 30 μg), rifampin (RIF, 30 μg),erythromycin (ERY, 10 μg), and trimethoprim/sulphamethoxazole (SXT, 1.25/23.75 μg). The inhibition zone diameters were interpreted according to the Clinical and Laboratory Standards Institute (CLSI) guidelines [46]. Isolates resistant to ≥3 different antimicrobial classes were classified as multidrug-resistant (MDR). The multiple antibiotic resistance (MAR) index for each isolate was calculated as the number of antimicrobials to which the isolate displayed resistance divided by the number of antimicrobials to which the isolate had been tested [47]. The minimum inhibitory concentration (MIC) of vancomycin (Sigma-Aldrich, Co., St. Louis, MO, USA) was estimated using the broth microdilution method following the CLSI guidelines [46], and the interpretive standards were those available in the relevant CLSI document. *C. pseudotuberculosis biovar Ovis* (ATCC 19410; Manassas, VA, USA) was used as a quality control.

### 2.5. Detection of Virulence and Resistance Genes

Plasmid DNA was extracted from bacterial strains using a Thermo Scientific GeneJET Plasmid Miniprep Kit (Thermo, Germany). PCR amplification of virulence-related genes of *C. pseudotuberculosis*, specifically *PIP*, *PLD*, *FagA*, *FagB*, *FagC*, and *FagD*, as well as antimicrobial resistance genes as *erm*(X) (macrolide, lincosamides and streptogramins resistance); *aph(3″)-Ib* (streptomycin resistance); *aph(3′)-Ic* (kanamycin resistance); *sul1*(sulfonamide resistance); *tet*(W) (tetracycline resistance); *bla* (beta-lactam resistance); *cat1* (chloramphenicol resistance); *dfrA1*(trimethoprim resistance); *vanA* (vancomycin resistance); *bcrA* (bacitracin resistance); *ermC* (clindamycin resistance); *rpoB* (rifampin resistance and *qnrA* (nalidixic acid resistance) were performed by conventional PCR assays using the oligonucleotide primer sequences presented in Table 1.

PCR was performed using a PTC-100 programmable Peltier-Effect thermal cycler (Caerphilly, UK). The final volume of the reaction mixture was adjusted to 25 μL, consisting of 12.5 μL of DreamTaq TM Green Master Mix (2X) (Fermentas, Waltham, MA, USA), 0.4 μL of 100 pmoL of each primer (Sigma, USA), 5 μL of template DNA, and nuclease-free water added up to 25 μL. The PCR products were separated by electrophoresis on a 1.5% agarose gel (Applichem GmbH, Darmstadt, Germany,), and the gel was photographed using a gel documentation framework (Alpha Innotech, Biometra, Germany). The data were analyzed with the GeneTools analysis software (SynGen, London, UK). *C. pseudotuberculosis* ATCC-19410 was used as a quality control.

### 2.6. ERIC Genotyping

To analyze the fingerprinting profiles of different *C. pseudotuberculosis*-positive isolates, ERIC-PCR was conducted using the ERIC primers (Table 1), as previously described [45]. The ERIC fingerprinting data was transformed into a binary code depending on the presence or absence of each band. Dendrograms were generated by the unweighted pair group method with an arithmetic average (UPGMA) and Ward’s hierarchical clustering routine. Cluster analysis and dendrogram construction were performed using the *hclust* function in the “factoextra” package. The similarity index (Jaccard/Tanimoto Coefficient and number of intersecting elements) between all samples was calculated using the online tool (https://planetcalc.com/1664/; accessed on 20 March 2022).

### 2.7. DNA Sequencing and Analysis

The amplified DNA products were purified using the QIAquick PCR purification kit (QIAGEN, Valencia, CA, USA), following the manufacturer’s instructions. Sequencing was performed in both directions using 16S Rrna primers and an automated sequencer (Macrogen Inc., Korea ABI 3730XL DNA analyzer). The DNA sequences were compared to those published in National Center for Biotechnology Information databases (NCBI, www.ncbi.nlm.nih.gov; accessed on 25 March 2022) using the Basic Local Alignment Search Tool (BLAST). Nucleotide sequence alignment was applied using the MEGA7 program (http://www.megasoftware.net; accessed on 1 April 2022) [48]. A phylogenetic tree was constructed using the neighbor-joining method [49] and Kimura’s two-parameter method [50] with 1000 bootstrap iterations. The nucleotide sequences generated in the study were deposited in GenBank under accession numbers OP550121–OP550130.

### 2.8. Data Analysis

The data were analyzed and visualized using R software (R Core Team, 2022; version 4.2.0). The Wilcoxon test was used to examine the variation in the MAR index of *C. pseudotuberculosis* isolated from sheep and goats. The “ggplot2” package was utilized for data visualization.

## 3. Results

### 3.1. Study Population

A total of 220 animals, comprising 130 sheep and 90 goats from 39 mixed smallholder flocks, were examined for clinical signs of CLA. The animals were raised in four districts of Sharkia Governorate, namely Zagazig (90 animals from 15 flocks), Belbeis (55 animals from 10 flocks), Abo-Hammad (40 animals from 8 flocks), and Menia Elkamh (35 animals from 6 flocks). The majority of the animals were males (58.6%), and 65% of them were fed concentrates (Table 2).

The animals were classified into three age groups: <1 year (18.2%), 1–2 years (37.7%), and >2 years (44.1%). During the clinical examination, it was found that 36 animals (16.4%) had skin lesions. The skin lesions were due to shearing, castration, docking, umbilical, and tagging wounds. Moreover, 54 animals (24.6%) had enlarged superficial lymph nodes in various parts of their bodies, while 21 animals (9.5%) had both skin lesions and enlarged lymph nodes. The abscesses were soft, relatively large, and contained cheesy purulent material that had a milky white or yellow-green color.

### 3.2. C. pseudotuberculosis Isolation and Identification

*C. pseudotuberculosis* was isolated in pure culture from 54 purulent samples (24 from sheep and 30 from goats). Morphological and biochemical characteristics were used to identify all bacterial isolates, which were Gram-positive pleomorphic organisms, appearing as coccoid to filamentous rods arranged in single or paired acute angles (resembling Chinese letters) when viewed under the ordinary microscope. Biochemical tests revealed that the isolates were catalase and urease-positive, and negative for the oxidase test, lactose, trehalose, xylose fermentation, esculin hydrolysis, and nitrate reduction, indicating that they belonged to biovar *Ovis*.

On blood agar, colonies were small, white, circular, and beta hemolytic after 48 h of incubation. The reverse CAMP test with *Staphylococcus aureus* was positive for all 54 *C. pseudotuberculosis* isolates. Furthermore, a zone of synergistic hemolysis appeared around the colonies of *C. pseudotuberculosis* when tested with *Rhodococcus equi*. Finally, confirmation of all bacterial isolates was achieved through 16S rRNA and RNA polymerase beta-subunit (*rpoB*) gene-specific PCR, which yielded PCR products of 845 and 446 base pairs, respectively.

### 3.3. Antimicrobial Resistance of C. pseudotuberculosis Isolates

The antimicrobial susceptibility profiles of the 54 *C. pseudotuberculosis* isolates against 24 antimicrobial agents are shown in Table 3. All isolates exhibited 100% resistance to bacitracin and florfenicol and susceptibility to norfloxacin. The majority of isolates exhibited high levels of resistance to penicillin, erythromycin (92.6% each), and cephradine (88.9%). Interestingly, 7.4% of *C. pseudotuberculosis* isolates were vancomycin-resistant, with minimum inhibitory concentration (MIC) values of ≥64.

Table 4 shows the frequency of *C. pseudotuberculosis* resistant isolates in both sheep and goats. All sheep isolates were 100% resistant to streptomycin, erythromycin, trimethoprim/sulphamethoxazole, bacitracin, and florfenicol but 100% susceptible to norfloxacin. However, goat’s isolates were all resistant to cephradine, bacitracinand florfenicolbut susceptible to ampicillin/sulbactam, norfloxacin, and vancomycin. Resistance to vancomycin was reported in four (16.7%) of the *C. pseudotuberculosis* recovered from sheep for the first time in Egypt.

All of the *C. pseudotuberculosis* isolates in this study showed MDR with an average MAR index of 0.63, ranging from 0.38 to 0.88. The highest MAR index (0.88) was found in an isolate recovered from a sheep. However, there was no statistically significant difference (*p* = 0.065) between the MAR index of isolates recovered from sheep and goats (Figure 2).

### 3.4. Virulence and Resistance Genes

The *PLD*, *PIP,* and *FagA* virulence genes were found in all (100%) of the *C. pseudotuberculosis* isolates recovered from both sheep and goats. However, the *FagB*, *FagC,* and *FagD* virulence genes were found in 24 (100%), 20 (83%), and 18 (75%) of the isolates recovered from sheep, respectively. From goats, *FagB* was found in 26 (87%), *FagC* in 26 (87%), and *FagD* in 18 (60%) of the isolates (Figure 3).

Figure 4 shows the frequency of antimicrobial resistance genes in *C. pseudotuberculosis* isolates recovered from both sheep and goats. The *β*-lactam resistance gene (*bla*) was present in all of the isolates, while the glycopeptide (*vanA*) resistance gene was only found in 8% of the isolates from sheep. Additionally, 83% of the isolates from sheep carried the aminoglycoside (*aph (3″)-lb*), chloramphenicol (*cat1*), and bacitracin (*bcrA*) resistance genes. For the isolates recovered from goats, 73% were found to contain macrolides (*ermX*), sulfonamide (*sul1*), and bacitracin (*bcrA*) resistance genes.

### 3.5. Genotyping and Phylogenetic Analysis

Ten MDR *C. pseudotuberculosis* isolates from sheep and goats (5 of each) were analyzed using ERIC-PCR fingerprinting and 16S rRNA sequencing. The ERIC-PCR dendrogram analysis showed that the 10 isolates were grouped into three clusters based on their similarity (Figure 5). Two of the clusters contained isolates from both sheep and goats, with a similarity index of 83.6%.

The 16S rRNA of 10 virulent MDR *C. pseudotuberculosis* isolates from sheep and goats was sequenced and uploaded to GenBank (accession numbers OP550121 to OP550130). In addition, two published sequences of biotypes equi (GenBank accession numbers CP003540.3 and CP003652.3) were included in the analysis to differentiate between *C. pseudotuberculosis* biovar *Ovis* and *Equi*. The phylogenetic tree showed that the 16S rRNA sequences of *C. pseudotuberculosis* isolates from sheep and goats clustered in the biovar *Ovis* lineage and were clearly differentiated from biovar *Equi* (Figure 6). Therefore, the nucleotide disparity did not alter protein expression. However, a *C. pseudotuberculosis* sequence from a sheep source (GenBank accession No. OP550123.1) showed sequence heterogeneity and was closely related to other published sequences.

## 4. Discussion

*C. pseudotuberculosis biovar Ovis* is the causative agent of CLA, which is a chronic contagious disease of sheep and goats, particularly in small breeders [11,32]. Information on the prevalence of CLA is limited in many countries as it is not a notifiable disease and often proceeds subclinically without any sign of abscesses [21]. In this study, CLA-compatible lesions were detected in 54 (24.54%) of the 220 studied animals, which were then confirmed by *C. pseudotuberculosis* isolation and subsequent PCR identification. The high infection rate may result from inadequate vaccination, sharing contaminated instruments, a dry climate, and an inefficient quarantine protocol adopted by small breeders. Our results were higher than the results reported by Oreiby and coauthors [2], who recorded a lower infection rate (6.7%) based on clinical lesions of CLA, with rates of 7.54% in sheep and 3.98% in goats.

In this study, a lower CLA infection rate was observed in sheep (18.46%) compared with goats (33.3%). Previous studies in Egypt have reported higher infection rates in sheep than in goats [2,6]. However, the isolation rate of *C. pseudotuberculosis* in this study was similar to the 25.66% recorded in goats with CLA in Egypt [6] and the 39.2% reported in southwestern China [8]. Other studies have reported a higher infection rate in goats in Nigeria (50%) and India (51.9%) [52,53]. The higher CLA infection rate in goats could be explained by the goats’ tendency to scratch themselves against hard objects, making them more susceptible to superficial injuries than sheep [54]. The infection rate in sheep in this study was similar to the 22.7% reported based on clinical signs and the 20.1% reported based on bacteriological examination [55]. Furthermore, other studies have reported lower rates (10.47% and 3.71%) in different Egyptian governorates [56,57]. The variability of CLA prevalence between studies may be attributed to management and environmental conditions, as well as the abundance of sheep and goat flocks within other localities. The present study involved sheep and goats from both fixed and mobile flocks. The lowest infection rate of 18.8% was observed in Zagazig, where most flocks are fixed, in contrast to other study localities where flocks are mobile. This finding is consistent with previous research by Selim and coworkers [57], who also reported that mobile flocks had a higher risk of infection compared to fixed flocks, attributing it to the fact that free-grazing animals in mobile flocks are more likely to come into contact with other infected animals.

It is worth mentioning that, in the present study, *C. pseudotuberculosis* was isolated from all purulent material samples collected from clinically infected animals and confirmed by 16S rRNA and RNA polymerase b-subunit (*rpoB*) gene-specific PCR. This result indicates that the sensitivities of bacterial culture and PCR were the same, suggesting the adoption of PCR as an alternative to cumbersome bacterial culture procedures. However, other studies have indicated that the incidence of CLA on a clinical basis has always been higher than on a bacteriological or molecular basis [28,58,59].

The 54 *C. pseudotuberculosis* isolates were tested for antimicrobial susceptibility and exhibited varying susceptibility patterns. However, the majority of these isolates demonstrated resistance to numerous of the 24 tested antimicrobials. Most previous studies on the antimicrobial susceptibility of *C. pseudotuberculosis* showed diverse prevalence and patterns [8,21,32,60], which has been attributed by some authors to regional variations in antimicrobial use, the availability of over-the-counter antibiotics without prescriptions, and levels of veterinary service provided [60]. In Egypt, antimicrobials are available over-the-counter and extensively used by smallholder breeders as prophylactics and/or growth promoters [22], which might explain the high resistance rate reported in this study.

*C. pseudotuberculosis* isolates recovered in the present study showed high rates of resistance to bacitracin, florfenicol, penicillin, and erythromycin, which is consistent with previous studies from Saudi Arabia [61] and Kosovo [62]. However, other studies have reported *C. pseudotuberculosis* isolates that were sensitive to the majority of antimicrobial classes [8,21,32] and attributed this to the limited use of antimicrobials in sheep and goats. It is worth mentioning that all (100%) of the *C. pseudotuberculosis* isolates recovered in the current study were sensitive to norfloxacin, while the emergence of vancomycin resistance was noted in 7.4% of the examined isolates as a first report in Egypt. This finding was similar to that of Abebe and Sisay Tessema [21], who revealed that all of their *C. pseudotuberculosis* isolates were 100% sensitive to norfloxacin and 20.3% were resistant to vancomycin.

In this study, the presence of MDR *C. pseudotuberculosis* was found to be high, with most of the isolates being resistant to clinically important antimicrobial classes including cephalosporins, aminoglycosides, and macrolides, which are listed by the WHO as critically important antimicrobials for human health [51]. Considering its zoonotic potential and economic impact, MDR *C. pseudotuberculosis* could pose a potential threat to public health.

Herein, multiple virulence genes (*PLD*, *PIP*, *FagA*, *FagB*, *FagC*, and *FagD*) were detected in *C. pseudotuberculosis* isolates, with a consistent virulence profile of *PLD*, *PIP*, and *FagA* identified in all isolates. Similar virulence profiles were also detected in *C. pseudotuberculosis* isolates recovered from superficial abscesses in previous studies [8,63]. However, variations in virulence profiles between studies suggested that *C. pseudotuberculosis* strains may have different pathogenic modes [32]. For instance, the virulence profiles of *PLD*, *FagA*, *FagB*, *FagC*, and *FagD* were more prevalent among *C. pseudotuberculosis* strains isolated from visceral abscesses compared to surface abscesses, indicating variations in the invasive potential of *C. pseudotuberculosis* strains [32].

Notably, *β*-lactam and aminoglycoside resistance genes were detected in most of our *C. pseudotuberculosis* isolates recovered from both sheep and goats. These results are not surprising as the isolates showed high resistance to *β*-lactams and aminoglycoside antimicrobial agents in susceptibility tests, which is consistent with previous findings in Egypt [56] and Argentina [64], although our results were higher. All resistance genes were detected in *C. pseudotuberculosis* isolates from both sheep and goats, except for the *vanA* gene, which was only found in sheep, suggesting that antimicrobial misuse may be more prevalent in sheep compared to goats. This variation could be attributed to local antimicrobial usage practices and farm biosecurity and management, as previous studies have reported an association between resistance genes and farm practices [65].

Various molecular methods, such as ERIC-PCR, restriction fragment length polymorphism (RFLP), and whole genome sequence analysis, have been used to classify and identify the genetic relationship between *C. pseudotuberculosis* isolates from sheep and goats [66,67]. In the current study, ERIC-PCR fingerprinting grouped the 10 MDR *C. pseudotuberculosis* isolates into three clusters, with high similarity between isolates from sheep and goats. Similarly, a recent study by Torky et al. [68] reported clusters of *C. pseudotuberculosis* isolates from sheep and goats in the same group, which could be attributed to the endemic nature of CLA in Egypt and the practice of raising sheep and goats together in the same flock. Furthermore, *C. pseudotuberculosis* has been shown to be highly persistent within the farm environment and can survive for up to 6 months in the environment, increasing the likelihood of transmission between species [69].

A phylogenetic analysis based on 16S rRNA sequence was used in the present study to differentiate between *C. pseudotuberculosis* biovar *Ovis* and *Equi*. However, other studies have reported that genes like *rpoB* [63,70] and *fusA* [8,71], which have higher polymorphism than 16S rRNA, are more effective for differentiation. In addition, our analysis revealed low bootstrap values and intergroup divergence within biotype *Ovis*, consistent with previous studies that reported genetic diversity within biotype *Ovis* is independent of host and geographic origins [8,71]. This could be due to the fact that this study was conducted on sheep and goats in a smallholder production system where close coexistence or even common grazing and housing among different flocks is possible [60].

An important limitation in this study is the sampling strategy, which involved collecting samples only from sheep and goats with superficial lymph node enlargement and only from a limited number of farmers who willingly agreed to take part. This approach may explain the different estimates recorded, but the results cannot be extrapolated to a wider national context. However, it is worth noting that the classical “external” form of CLA is the most prevalent, while abscessation in internal lymph nodes and organs occurs less frequently [5].

## 5. Conclusions

This study revealed that superficial abscesses in sheep and goats, raised under smallholder production systems in four districts in Egypt, were mostly associated with *C. pseudotuberculosis* infection. *C. pseudotuberculosis* was isolated from both sheep and goats, with all isolates belonging to *C. pseudotuberculosis biovar Ovis*. Furthermore, *C. pseudotuberculosis* exhibited high levels of resistance to multiple antimicrobial agents, as well as varying patterns of virulence and resistance genes. Therefore, to effectively control and reduce the impact of *C. pseudotuberculosis* infection, a sustainable control strategy and increased knowledge and awareness among smallholder breeders are recommended.

## Figures and Tables

**Figure 1 animals-13-02337-f001:**
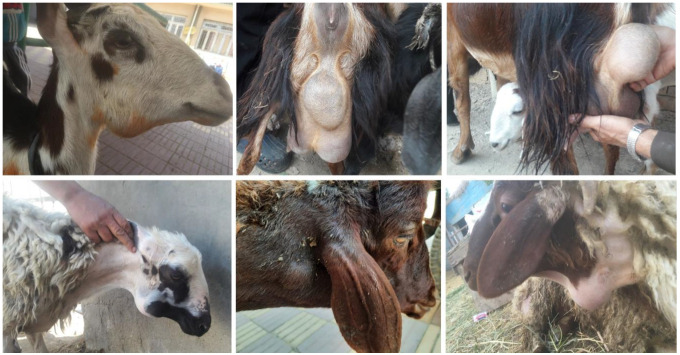
Abscess formation in goats (**upper row**) and sheep (**lower row**) occurs in different body locations due to infection with *C. pseudotuberculosis*.

**Figure 2 animals-13-02337-f002:**
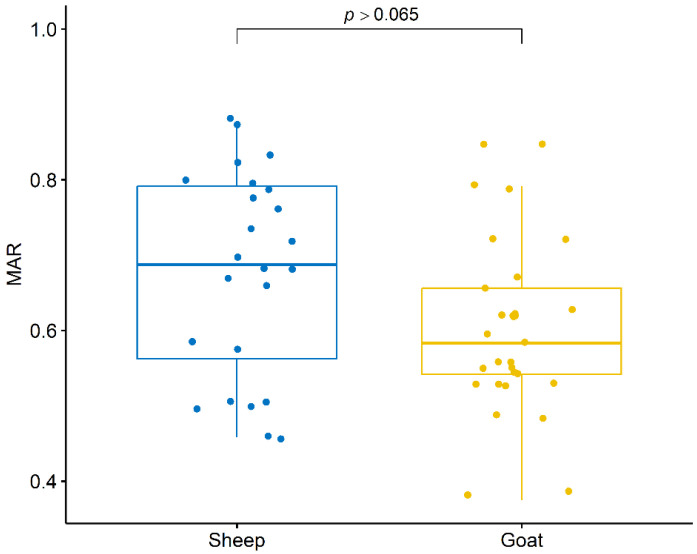
Multiple Antimicrobial Resistance (MAR) Index of *C. pseudotuberculosis* isolates recovered from sheep and goats with caseous lymphadenitis.

**Figure 3 animals-13-02337-f003:**
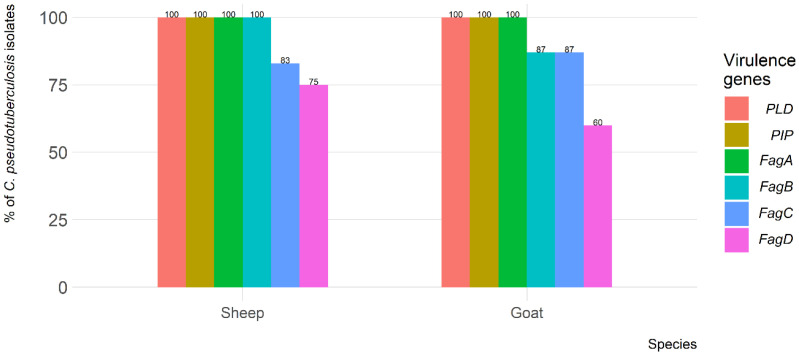
Frequency of virulence genes of *C. pseudotuberculosis* isolates recovered from sheep and goats with caseous lymphadenitis.

**Figure 4 animals-13-02337-f004:**
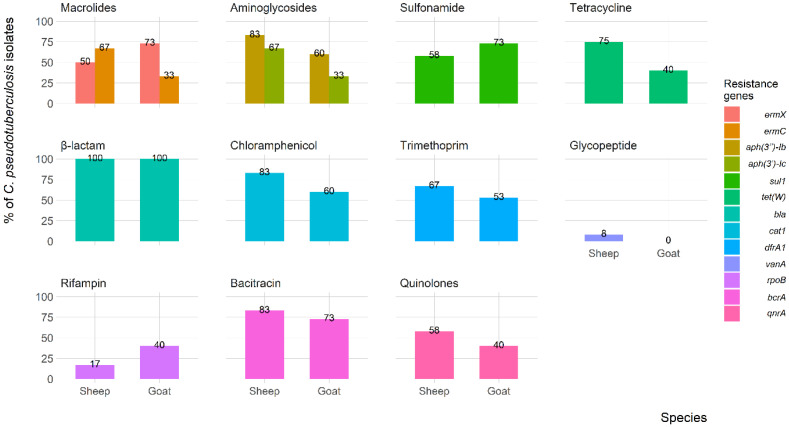
Frequency of antimicrobial resistance genes of *C. pseudotuberculosis* isolates recovered from sheep and goats with caseous lymphadenitis.

**Figure 5 animals-13-02337-f005:**
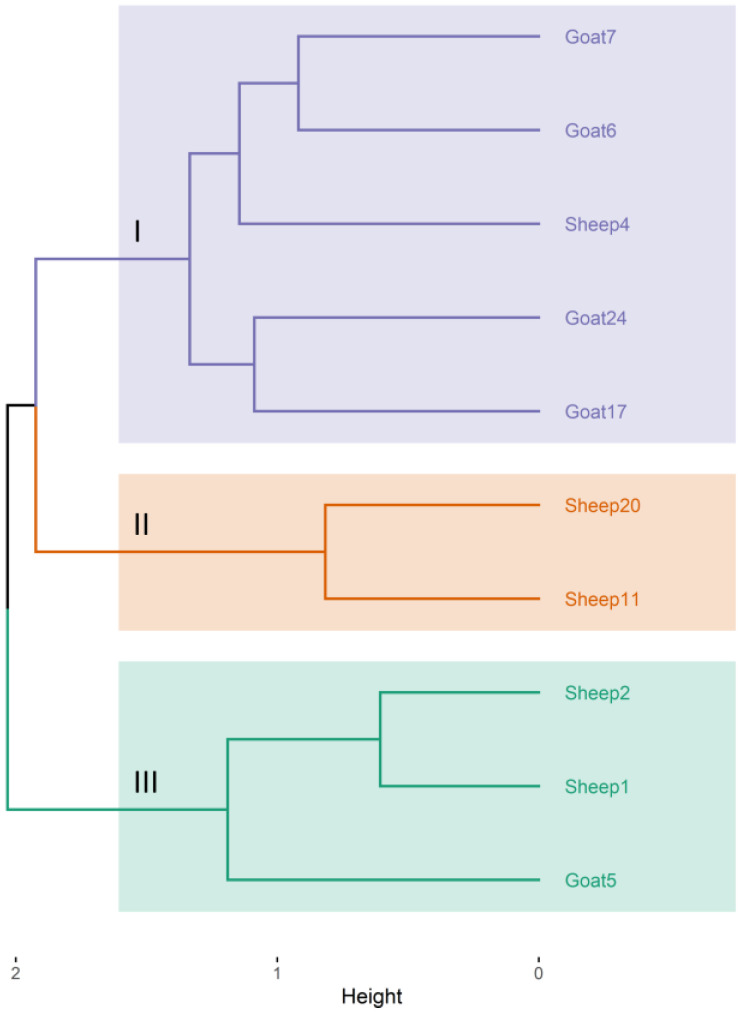
Dendrogram based on ERIC-PCR genotyping of *C. pseudotuberculosis* isolates (*n* = 10) recovered from sheep and goats with caseous lymphadenitis. Different colors corresponded to the three clusters identified.

**Figure 6 animals-13-02337-f006:**
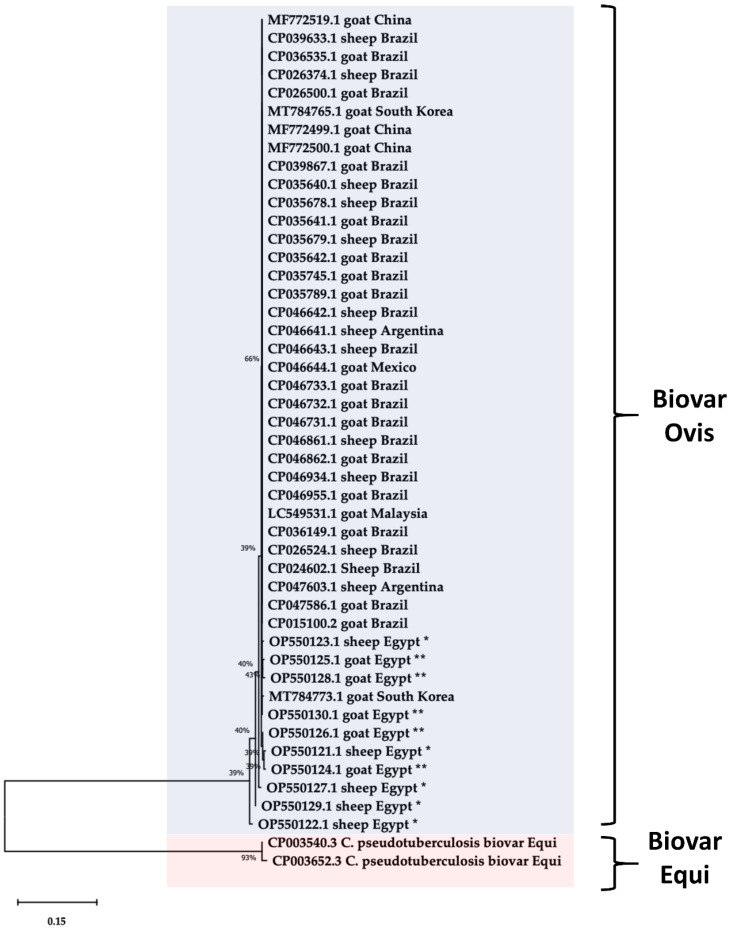
Phylogenetic tree of *C. pseudotuberculosis* isolated from sheep (*) and goats (**) based on the sequence of 16S rRNA. A phylogenetic tree was generated using the neighbor-joining method and 1000 bootstrap values.

**Table 1 animals-13-02337-t001:** Oligonucleotide primer sequences used in this study.

Target Genes	Nucleotide Sequence (5′→3′)	Amplicon Size (bp)	Annealing Temperature (°C)	Reference
Bacterial identification:				
16S rRNA	F: ACCGCACTTTAGTGTGTGTG	815	56	[28]
	R: TCTCTACGCCGATCTTGTAT			
*rpoB*	F: CGTATGAACATCGGCCAGGT	446	55	[29]
	R: TCCATTTCGCCGAAGCGCTG			
Virulence genes:				
*PLD*	F: ATAAGCGTAAGCAGGGAGCA	203	56	[30]
	R: ATCAGCGGTGATTGTCTTCCAGG			
*PIP*	F: AACTGCGGCTTTCTTTATTC	551	54	[31]
	R: GACAAGTGGGAACGGTATCT			
*FagA*	F:AGCAAGACCAAGAGACATGC	245	58	[32]
	R:AGTCTCAGCCCAACGTACAG			
*FagB*	F:GTGAGAAGAACCCCGGTATAAG	291	55	[32]
	R: TACCGCACTTATTCTGACACTG			
*FagC*	F:GTTTGGCTATCTCCTTGGTATG	173	60	[32]
	R: CGACCTTAGTGTTGACATACCC			
*FagD*	F:GAGACTATCGACCAGGCAG	226	61	[32]
	R: ACTTCTTGGGGAGCAGTTCT			
Resistance genes:				
*ermX*	F:TCCTTACCAGTGCCCTTATCC	390	65	[33]
	R: GAGTTCCAGCGCATCACC			
*ermC*	ermC-106: ATTGTGGATCGGGCAAATATT	447	53	[34]
	ermC-535: TGGAGGGGGAGAAAAATG			
*vanA*	F:GGGAAAACGACAATTGC	732	60	[35]
	R: GTACAATGCGGCCGTTA			
*rpoB*	TbRif1: AG ACG TTG ATC AAC ATC CG	304	55	[36]
	TbRif2: TAC GGC GTT TCG ATG AAC			
*bcrA*	CP-F: GGCAATACCAAGCCGTTGCTTCAT	408	55	[37]
	CP-R: TTACGAAGCGATACGGAACAGCCA			
*qnrA*	QP1: GATAAAGTTTTTCAGCAAGAGG	657	57	[38]
	QP2: ATCCAGATCGGCAAAGGTTA			
*ermA*	F: TCCTTACCAGTGCCCTTATCC	390	65	[33]
	R: GAGTTCCAGCGCATCACC			
*aph(3″)-Ib*	F: CTTGGTGATAACGGCAATTC	548	52	[39]
	R: CCAATCGCAGATAGAAGGC			
*aph(3′)-Ic*	F: CGAGCATCAAATGAAACTGC	624	54	[40]
	R: GCGTTGCCAATGATGTTACAG			
*sul1*	F: CGG CGT GGG CTA CCT GAA CG	433	50	[41]
	R: GCC GAT CGC GTG AAG TTC CG			
*tet* (W)	F: TTCGATGGTGGCACAGTA	234	60	[42]
	R: TTGTTCGGCTGGAACGTA			
*bla*	F: CAGTCTAGCCACTTCGCCAAT	808	55	[43]
	R: TGACTGCACGGATGGAGATGG			
*cat1*	F: AGTTGCTCAATGTACCTATAACC	547	55	[44]
	R: TTGTAATTCATTAAGCATTCTGCC			
*dfrA1*	F: CTCACGATAAACAAAGAGTCA	201	50	[44]
	R: CAATCATTGCTTCGTATAACG			
ERIC-PCR:				
ERIC-1 ERIC-2	F: ATGTAAGCTCCTGGGGATTCAC	Variable	52	[45]
	R: AAGTAAGTGACTGGGGTGAGCG			

**Table 2 animals-13-02337-t002:** Study population demographic characteristics.

Variable	Categories	N (%)	Number of Examined (%)	
			Sheep	Goats
Gender				
	Female	91 (41.4)	57 (43.8)	34 (37.8)
	Male	129 (58.6)	73 (56.2)	56 (62.2)
Age				
	<1 year	40 (18.2)	25 (19.2)	15 (16.7)
	1–2 years	83 (37.7)	39 (30.0)	44 (48.9)
	>2 years	97 (44.1)	66 (50.8)	31 (34.4)
Locality				
	Zagazig	90 (40.9)	54 (41.5)	36 (40.0)
	Belbis	55 (25.0)	34 (26.1)	21 (23.3)
	Abo-Hammad	40 (18.2)	18 (13.9)	22 (24.4)
	Menia ElKamh	35 (15.9)	24 (18.5)	11 (12.3)
Feeding				
	Concentrates	143 (65.0)	83 (63.9)	60 (66.7)
	Grazing	77 (35.0)	47 (36.2)	30 (33.3)
Season				
	Autumn	40 (18.2)	22 (16.9)	18 (20.0)
	Spring	91 (41.4)	55 (42.3)	36 (40.0)
	Summer	33 (15.0)	18 (13.9)	15 (16.7)
	Winter	56 (25.4)	35 (26.9)	21 (23.3)
Skin lesions				
	Yes	36 (16.4)	19 (14.6)	17 (18.9)
	No	184 (83.6)	111 (85.4)	73 (81.1)
Enlarged lymph nodes				
	Yes	54 (24.6)	24 (18.5)	30 (33.3)
	No	166 (75.4)	106 (81.5)	60 (66.7)
Total		220	130	90

**Table 3 animals-13-02337-t003:** Source, antimicrobial resistance profiles, virulence, and resistance genes of *C. pseudotuberculosis* isolates recovered from sheep and goats with caseous lymphadenitis.

Isolate	Source	Antimicrobial Resistance Profiles ^1^	MAR Index ^2^	Virulence Genes	Resistance Genes
S1	Sheep	PEN, NB, AMC, AM, OXA, VAN, CRO, CXM, CEP, BAC, AMP, DOX, CLI, NAL, STR, KAN, AMK, NEO, FFC, ERY, SXT	0.88	*PLD, PIP, FagA, FagB, FagC, FagD*	*aph(3″)-Ib, aph(3′)-Ic, sul1, tet(W), bla, cat1, dfrA1, ermC, vanA, bcrA, qnrA*
S2	Sheep	PEN, NB, AMC, AM, OXA, VAN, CRO, CXM, BAC, AMP, DOX, CLI, NAL, STR, KAN, AMK, NEO, FFC, ERY, SXT	0.83	*PLD, PIP, FagA, FagB, FagC, FagD*	*aph(3″)-Ib, aph(3′)-Ic, tet(W), bla, cat1, dfrA1, ermC, vanA, bcrA, qnrA*
S3	Sheep	PEN, NB, AMC, AM, OXA, BAC, DOX, NAL, STR, NEO, FFC, ERY, SXT	0.5	*PLD, PIP, FagA, FagB, FagC, FagD*	*aph(3″)-Ib, tet(W), bla, cat1, dfrA1*
S4	Sheep	PEN, NB, AMC, OXA, BAC, DOX, CLI, NAL, STR, KAN, NEO, FFC, ERY, SXT	0.58	*PLD, PIP, FagA, FagB, FagC, FagD*	*aph(3″)-Ib, aph(3′)-Ic, tet(W), bla, cat1, dfrA1, ermC, bcrA, qnrA*
S5	Goat	PEN, NB, AMC, OXA, CEP, BAC, DOX, CLI, NAL, STR, KAN, NEO, FFC, ERY, SXT	0.54	*PLD, PIP, FagA, FagB, FagC, FagD*	*aph(3″)-Ib, aph(3′)-Ic, tet(W), bla, cat1, dfrA1, ermC, bcrA, qnrA*
S6	Goat	PEN, NB, AMC, OXA, CEP, BAC, DOX, CLI, NAL, STR, KAN, NEO, FFC, ERY, SXT	0.58	*PLD, PIP, FagA, FagB, FagC, FagD*	*ermX, aph(3″)-Ib, aph(3′)-Ic, sul1, tet(W), bla, cat1, ermC, bcrA, qnrA*
S7	Goat	PEN, NB, AMC, OXA, PIP, CEP, BAC, DOX, CLI, NAL, STR, KAN, FFC, ERY, SXT	0.54	*PLD, PIP, FagA, FagB, FagC, FagD*	*sul1, tet(W), bla, cat1, qnrA*
S8	Sheep	PEN, NB, AMC, OXA, PIP, CRO, CXM, CEP, BAC, AMP, DOX, CLI, NAL, STR, KAN, FFC, ERY, SXT	0.63	*PLD, PIP, FagA, FagB, FagC*	*aph(3″)-Ib, aph(3′)-Ic, sul1, tet(W), bla, cat1, ermC, bcrA, qnrA*
S9	Sheep	PEN, NB, AMC, OXA, PIP, CEP, BAC, AMP, DOX, CLI, NAL, STR, KAN, FFC, ERY, SXT	0.63	*PLD, PIP, FagA, FagB, FagC, FagD*	*aph(3″)-Ib, aph(3′)-Ic, sul1, tet(W), bla, ermC, bcrA*
S10	Goat	PEN, AMC, OXA, PIP, CXM, CEP, BAC, AMP, DOX, CLI, FFC, ERY, SXT	0.75	*PLD, PIP, FagA, FagB, FagC, FagD*	*ermX, aph(3′)-Ic, sul1, tet(W), bla, ermC, bcrA, qnrA*
S11	Sheep	PEN, NB, AMC, OXA, PIP, CXM, CEP, BAC, AMP, DOX, CLI, NAL, STR, KAN, NEO, FFC, ERY, SXT	0.63	*PLD, PIP, FagA, FagB, FagC, FagD*	*ermX, sul1, bla, dfrA1, ermC, bcrA, qnrA*
S12	Goat	PEN, AMC, OXA, PIP, CEP, BAC, AMP, DOX, STR, KAN, NEO, FFC	0.5	*PLD, PIP, FagA, FagB, FagC, FagD*	*sul1, bla, dfrA1*
S13	Goat	PEN, NB, AMC, OXA, PIP, CRO, CXM, CEP, BAC, AMP, CLI, STR, KAN, AMK, NEO, FFC, RIF, ERY, SXT	0.79	*PLD, PIP, FagA, FagB, FagC, FagD*	*ermX, aph(3″)-Ib, aph(3′)-Ic, bla, rpoB*
S14	Goat	PEN, NB, AMC, OXA, PIP, CRO, CXM, CEP, BAC, AMP, DOX, CLI, STR, KAN, AMK, NEO, FFC, RIF, ERY, SXT	0.83	*PLD, PIP, FagA, FagB, FagC, FagD*	*ermX, aph(3″)-Ib, sul1, tet(W), bla, cat1, dfrA1, ermC, rpoB, bcrA*
S15	Sheep	PEN, NB, OXA, PIP, CRO, CXM, CEP, BAC, AMP, DOX, CLI, NAL, STR, KAN, AMK, FFC, RIF, ERY, SXT	0.71	*PLD, PIP, FagA, FagB, FagC, FagD*	*ermX, aph(3″)-Ib, aph(3′)-Ic, tet(W), bla, cat1, dfrA1, ermC, rpoB, bcrA, qnrA*
S16	Sheep	PEN, NB, AMC, OXA, PIP, CRO, CXM, CEP, BAC, AMP, DOX, CLI, NAL, STR, KAN, AMK, FFC, ERY, SXT	0.67	*PLD, PIP, FagA, FagB, FagC, FagD*	*ermX, aph(3″)-Ib, aph(3′)-Ic, tet(W), bla, cat1, dfrA1, ermC, bcrA, qnrA*
S17	Goat	PEN, NB, OXA, PIP, CRO, CXM, CEP, BAC, CLI, NAL, STR, KAN, AMK, NEO, FFC, ERY, SXT	0.79	*PLD, PIP, FagA, FagB, FagC, FagD*	*ermX, sul1, bla, cat1, dfrA1, bcrA*
S18	Sheep	PEN, AM, OXA, PIP, CRO, CXM, CEP, BAC, STR, KAN, AMK, NEO, FFC, RIF, ERY, SXT	0.54	*PLD, PIP, FagA, FagB, FagC, FagD*	*ermX, sul1, bla, cat1, dfrA1, bcrA*
S19	Goat	PEN, PIP, CXM, CEP, BAC, DOX, AMK, FFC, SXT	0.79	*PLD, PIP, FagA, FagB, FagC, FagD*	*sul1, bla, cat1, dfrA1, rpoB*
S20	Sheep	PIP, CXM, CEP, BAC, DOX, STR, KAN, FFC, RIF, ERY, SXT	0.75	*PLD, PIP, FagA, FagB*	*ermX, aph(3″)-Ib, aph(3′)-Ic, sul1, tet(W), bla, cat1, rpoB, bcrA*
S21	Sheep	PEN, AMC, CXM, CEP, BAC, AMP, DOX, STR, KAN, FFC, ERY, SXT	0.5	*PLD, PIP, FagA, FagB*	*ermX, aph(3″)-Ib, sul1, bla, cat1*
S22	Goat	PEN, PIP, CXM, CEP, BAC, DOX, STR, KAN, NEO, FFC, RIF, ERY, SXT	0.71	*PLD, PIP, FagA, FagB*	*ermX, aph(3″)-Ib, aph(3′)-Ic, sul1, bla, cat1, rpoB, bcrA*
S23	Goat	NB, OXA, PIP, CXM, CEP, BAC, STR, KAN, NEO, FFC, RIF, ERY, SXT	0.67	*PLD, PIP, FagA, FagB, FagC*	*ermX, aph(3″)-Ib, sul1, bla, cat1, dfrA1, bcrA*
S24	Goat	PEN, NB, PIP, CXM, CEP, BAC, AMP, DOX, CLI, NAL, STR, NEO, FFC, RIF, ERY, SXT	0.38	*PLD, PIP, FagA, FagC*	*ermX, aph(3″)-Ib, sul1, tet(W), bla, dfrA1, ermC, rpoB, bcrA, qnrA*
S25	Goat	PEN, PIP, CXM, CEP, BAC, AMP, DOX, STR, KAN, NEO, FFC, ERY, SXT	0.46	*PLD, PIP, FagA, FagB*	*ermX, aph(3″)-Ib, sul1, bla, dfrA1, bcrA*
S26	Goat	PEN, NB, OXA, PIP, CXM, CEP, BAC, AMP, DOX, STR, NEO, FFC, RIF, ERY	0.5	*PLD, PIP, FagA, FagB, FagC*	*ermX, aph(3″)-Ib, bla, cat1, rpoB, bcrA*
S27	Goat	PEN, NB, OXA, PIP, CXM, CEP, BAC, AMP, DOX, NAL, NEO, FFC, ERY	0.54	*PLD, PIP, FagA, FagC*	*ermX, bla, bcrA, qnrA*
S28	Sheep	PEN, NB, AMC, OXA, PIP, CXM, CEP, BAC, AMP, DOX, CLI, NAL, STR, KAN, NEO, FFC, ERY, SXT	0.79	*PLD, PIP, FagA, FagB, FagC, FagD*	*ermX, sul1, bla, dfrA1, ermC, bcrA, qnrA*
S29	Goat	PEN, AMC, OXA, PIP, CEP, BAC, AMP, DOX, STR, KAN, NEO, FFC	0.54	*PLD, PIP, FagA, FagB, FagC, FagD*	*sul1, bla, dfrA1*
S30	Goat	PEN, NB, AMC, OXA, PIP, CRO, CXM, CEP, BAC, AMP, CLI, STR, KAN, AMK, NEO, FFC, RIF, ERY, SXT	0.67	*PLD, PIP, FagA, FagB, FagC, FagD*	*ermX, aph(3″)-Ib, aph(3′)-Ic, bla, rpoB*
S31	Goat	PEN, NB, AMC, OXA, PIP, CRO, CXM, CEP, BAC, AMP, DOX, CLI, STR, KAN, AMK, NEO, FFC, RIF, ERY, SXT	0.54	*PLD, PIP, FagA, FagB, FagC, FagD*	*ermX, aph(3″)-Ib, sul1, tet(W), bla, cat1, dfrA1, ermC, rpoB, bcrA*
S32	Sheep	PEN, NB, OXA, PIP, CRO, CXM, CEP, BAC, AMP, DOX, CLI, NAL, STR, KAN, AMK, FFC, RIF, ERY, SXT	0.83	*PLD, PIP, FagA, FagB, FagC, FagD*	*ermX, aph(3″)-Ib, aph(3′)-Ic, tet(W), bla, cat1, dfrA1, ermC, rpoB, bcrA, qnrA*
S33	Sheep	PEN, NB, AMC, OXA, PIP, CRO, CXM, CEP, BAC, AMP, DOX, CLI, NAL, STR, KAN, AMK, FFC, ERY, SXT	0.79	*PLD, PIP, FagA, FagB, FagC, FagD*	*ermX, aph(3″)-Ib, aph(3′)-Ic, tet(W), bla, cat1, dfrA1, ermC, bcrA, qnrA*
S34	Goat	PEN, NB, OXA, PIP, CRO, CXM, CEP, BAC, CLI, NAL, STR, KAN, AMK, NEO, FFC, ERY, SXT	0.58	*PLD, PIP, FagA, FagB, FagC, FagD*	*ermX, sul1, bla, cat1, dfrA1, bcrA*
S35	Sheep	PEN, AM, OXA, PIP, CRO, CXM, CEP, BAC, STR, KAN, AMK, NEO, FFC, RIF, ERY, SXT	0.79	*PLD, PIP, FagA, FagB, FagC, FagD*	*ermX, sul1, bla, cat1, dfrA1, bcrA*
S36	Goat	PEN, PIP, CXM, CEP, BAC, DOX, AMK, FFC, SXT	0.54	*PLD, PIP, FagA, FagB, FagC, FagD*	*sul1, bla, cat1, dfrA1, rpoB*
S37	Sheep	PIP, CXM, CEP, BAC, DOX, STR, KAN, FFC, RIF, ERY, SXT	0.71	*PLD, PIP, FagA, FagB*	*ermX, aph(3″)-Ib, aph(3′)-Ic, sul1, tet(W), bla, cat1, rpoB, bcrA*
S38	Sheep	PEN, AMC, CXM, CEP, BAC, AMP, DOX, STR, KAN, FFC, ERY, SXT	0.67	*PLD, PIP, FagA, FagB*	*ermX, aph(3″)-Ib, sul1, bla, cat1*
S39	Goat	PEN, PIP, CXM, CEP, BAC, DOX, STR, KAN, NEO, FFC, RIF, ERY, SXT	0.88	*PLD, PIP, FagA, FagB*	*ermX, aph(3″)-Ib, aph(3′)-Ic, sul1, bla, cat1, rpoB, bcrA*
S40	Goat	NB, OXA, PIP, CXM, CEP, BAC, STR, KAN, NEO, FFC, RIF, ERY, SXT	0.83	*PLD, PIP, FagA, FagB, FagC*	*ermX, aph(3″)-Ib, sul1, bla, cat1, dfrA1, bcrA*
S41	Goat	PEN, NB, PIP, CXM, CEP, BAC, AMP, DOX, CLI, NAL, STR, NEO, FFC, RIF, ERY, SXT	0.5	*PLD, PIP, FagA, FagC*	*ermX, aph(3″)-Ib, sul1, tet(W), bla, dfrA1, ermC, rpoB, bcrA, qnrA*
S42	Goat	PEN, PIP, CXM, CEP, BAC, AMP, DOX, STR, KAN, NEO, FFC, ERY, SXT	0.58	*PLD, PIP, FagA, FagB*	*ermX, aph(3″)-Ib, sul1, bla, dfrA1, bcrA*
S43	Goat	PEN, NB, OXA, PIP, CXM, CEP, BAC, AMP, DOX, STR, NEO, FFC, RIF, ERY	0.63	*PLD, PIP, FagA, FagB, FagC*	*ermX, aph(3″)-Ib, bla, cat1, rpoB, bcrA*
S44	Goat	PEN, NB, OXA, PIP, CXM, CEP, BAC, AMP, DOX, NAL, NEO, FFC, ERY	0.63	*PLD, PIP, FagA, FagC*	*ermX, bla, bcrA, qnrA*
S45	Sheep	PEN, NB, AMC, AM, OXA, VAN, CRO, CXM, CEP, BAC, AMP, DOX, CLI, NAL, STR, KAN, AMK, NEO, FFC, ERY, SXT	0.38	*PLD, PIP, FagA, FagB, FagC, FagD*	*aph(3″)-Ib, aph(3′)-Ic, sul1, tet(W), bla, cat1, dfrA1, ermC, bcrA, qnrA*
S46	Sheep	PEN, NB, AMC, AM, OXA, VAN, CRO, CXM, BAC, AMP, DOX, CLI, NAL, STR, KAN, AMK, NEO, FFC, ERY, SXT	0.46	*PLD, PIP, FagA, FagB, FagC, FagD*	*aph(3″)-Ib, aph(3′)-Ic, tet(W), bla, cat1, dfrA1, ermC, bcrA, qnrA*
S47	Sheep	PEN, NB, AMC, AM, OXA, BAC, DOX, NAL, STR, NEO, FFC, ERY, SXT	0.5	*PLD, PIP, FagA, FagB, FagC, FagD*	*aph(3″)-Ib, tet(W), bla, cat1, dfrA1*
S48	Sheep	PEN, NB, AMC, OXA, BAC, DOX, CLI, NAL, STR, KAN, NEO, FFC, ERY, SXT	0.54	*PLD, PIP, FagA, FagB, FagC, FagD*	*aph(3″)-Ib, aph(3′)-Ic, tet(W), bla, cat1, dfrA1, ermC, bcrA, qnrA*
S49	Goat	PEN, NB, AMC, OXA, CEP, BAC, DOX, CLI, NAL, STR, KAN, NEO, FFC, ERY, SXT	0.63	*PLD, PIP, FagA, FagB, FagC, FagD*	*aph(3″)-Ib, aph(3′)-Ic, tet(W), bla, cat1, dfrA1, ermC, bcrA, qnrA*
S50	Goat	PEN, NB, AMC, OXA, CEP, BAC, DOX, CLI, NAL, STR, KAN, NEO, FFC, ERY, SXT	0.71	*PLD, PIP, FagA, FagB, FagC, FagD*	*ermX, aph(3″)-Ib, aph(3′)-Ic, sul1, tet(W), bla, cat1, ermC, bcrA, qnrA*
S51	Goat	PEN, NB, AMC, OXA, PIP, CEP, BAC, DOX, CLI, NAL, STR, KAN, FFC, ERY, SXT	0.05	*PLD, PIP, FagA, FagB, FagC, FagD*	*sul1, tet(W), bla, cat1, qnrA*
S52	Sheep	PEN, NB, AMC, OXA, PIP, CRO, CXM, CEP, BAC, AMP, DOX, CLI, NAL, STR, KAN, FFC, ERY, SXT	0.54	*PLD, PIP, FagA, FagB, FagC*	*aph(3″)-Ib, aph(3′)-Ic, sul1, tet(W), bla, cat1, ermC, bcrA, qnrA*
S53	Sheep	PEN, NB, AMC, OXA, PIP, CEP, BAC, AMP, DOX, CLI, NAL, STR, KAN, FFC, ERY, SXT	0.67	*PLD, PIP, FagA, FagB, FagC, FagD*	*aph(3″)-Ib, aph(3′)-Ic, sul1, tet(W), bla, ermC, bcrA*
S54	Goat	PEN, AMC, OXA, PIP, CXM, CEP, BAC, AMP, DOX, CLI, FFC, ERY, SXT	0.54	*PLD, PIP, FagA, FagB, FagC, FagD*	*ermX, aph(3′)-Ic, sul1, tet(W), bla, ermC, bcrA, qnrA*

^1^ PEN: penicillin G; AMC: amoxicillin/clavulanic acid; OXA: oxacillin; SAM: ampicillin/sulbactam; AMP: ampicillin; STR: streptomycin; AMK: amikacin; PIP: piperacillin; CRO: ceftriaxone; CXM: cefuroxime sodium; CEP: cephradine; NAL: nalidixic acid; NOR: norfloxacin; KAN: kanamycin; NEO: neomycin; DOX: doxycycline; CLI: clindamycin; VAN: vancomycin; ERY: erythromycin; SXT: trimethoprim/sulphamethoxazole; FFC: florfenicol; NB: novobiocin; BAC: bacitracin; RIF: rifampin.^2^ MAR index: multiple antibiotic resistance (MAR) index.

**Table 4 animals-13-02337-t004:** Frequency of the antimicrobial resistant *C. pseudotuberculosis* isolates recovered from sheep and goats with caseous lymphadenitis.

Rank ^1^	Class	Agent ^2^	No. of Resistant *C. pseudotuberculosis* Isolates (%)		
			*n* = 54	Sheep (*n* = 24)	Goats (*n* = 30)
I	Aminoglycosides	AMK	18 (33.3)	10 (41.7)	8 (26.7)
		KAN	42 (77.8)	22 (91.7)	20 (66.7)
		NEO	36 (66.7)	12 (50.0)	24 (80.0)
		STR	48 (88.9)	24 (100)	24 (80.0)
I	Cephalosporins	CRO	18 (33.3)	12 (50.0)	6 (20.0)
		CXM	40 (74.1)	18 (75.0)	22 (73.3)
		CEP	48 (88.9)	18 (75.0)	30 (100)
I	Macrolides	ERY	50 (92.6)	24 (100)	26 (86.7)
I	Quinolones	NAL	30 (55.6)	18 (75.0)	12 (40.0)
		NOR	0 (0.0)	0 (0.0)	0 (0.0)
II	Aminocoumarins	NB	38 (70.4)	18 (75.0)	20 (66.7)
II	Glycopeptides	VAN	4 (7.4)	4 (16.7)	0 (0.0)
	Lincosamides	CLI	32 (59.3)	16 (66.7)	16 (53.3)
II	Penicillins	PEN	50 (92.6)	22 (91.7)	28 (93.3)
		AMC	32 (59.3)	18 (75.0)	14 (46.7)
		OXA	42 (77.8)	20 (83.3)	22 (73.3)
		SAM	8 (14.8)	8 (33.3)	0 (0.0)
		AMP	32 (59.3)	16 (66.7)	16 (53.3)
		PIP	40 (74.1)	14 (58.3)	26 (86.7)
II	Sulfonamides	SXT	48 (88.9)	24 (100)	24 (80.0)
II	Tetracyclines	DOX	46 (85.2)	22 (91.7)	24 (80.0)
	Amphenicols	FFC	54 (100)	24 (100)	30 (100)
	Bacitracin	BAC	54 (100)	24 (100)	30 (100)
	Rifampin	RIF	18 (33.3)	6 (25.0)	12 (40.0)

^1^ Rank I, critically important; rank II, highly important (based on World Health Organization’s categorization [51]).^2^ PEN: penicillin G; AMC: amoxicillin/clavulanic acid; OXA:oxacillin; SAM: ampicillin/sulbactam; AMP: ampicillin; STR: streptomycin; AMK: amikacin; PIP: piperacillin; CRO:ceftriaxone; CXM: cefuroxime sodium; CEP: cephradine; NAL: nalidixic acid; NOR: norfloxacin; KAN: kanamycin; NEO: neomycin; DOX: doxycycline; CLI: clindamycin; VAN: vancomycin; ERY: erythromycin; SXT: trimethoprim/sulphamethoxazole; FFC: florfenicol; NB: novobiocin; BAC: bacitracin; RIF: rifampin.

## Data Availability

The data presented in this study are available on request from the corresponding author.

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
