# Peer review of "Molecular Characterization and Antimicrobial Susceptibilities of Corynebacterium pseudotuberculosis Isolated from Caseous Lymphadenitis of Smallholder Sheep and Goats"

_animals, 2023, doi:10.3390/ani13142337_

Round 1

Reviewer 1 Report

Summary 

This paper describes detailed characterisation of Corynebacterium pseudotuberculosis isolates collected from cases of caseous lymphadentitis diagnosed in goats and sheep from smallholder herds/flocks in a region of Egypt.  A range of molecular techniques are employed.  In addition antibiotic sensitivity testing is also performed on these different isolates.  The results of this work are well presented and well structured in general. 

General concept comments: 

Article: In my opinion (and within the bounds of my experience) the description of the techniques employed is robust. The tables and figures are well constructed and easy to understand. I particularly liked the photographic images used here.

Specific comments/suggested edits:

Line 6 – the numeral 5 has been added to the second name on this line

Line 28 – elsewhere the paper refers to a total of 220 animals being examined, not 200 as here. 

Lines 41-43 – I do not feel the final line of the summary is justified by what goes before. 

Line 47 – here and elsewhere ‘resistance genes’ not resistant genes

Line 68 – 72 – as above, in my opinion, without describing the method by which the smallholder herds/flocks were selected for examination, what proportion of the herd/flock was examined etc. it is impossible to draw any wider conclusions on disease prevalence.  Without this context, I am not persuaded that the findings, interesting as they may be, ‘emphasized the necessity of conducting monitoring and surveillance of CLA in the flocks of smallholder farmers.’

Line 91 – C. pseudotuberculosis does not always infect humans should be changed.  Perhaps to ‘C. pseudotuberculosis may on rare occasions infect humans…’

Line 107 – This reference is incorrect as the article cited makes no reference to virulence genes.  A check by the authors that all referencing within the paper is correct, may be warranted.

Line 114 – change are to ‘is’  

2. Materials & Methods

Line 132 – This is described as a ‘cross-sectional study’ with no further detail provided. A description of how study herds/flocks were selected is required.  In addition as only 5-6 animals on average were examined at each site – was this a sub-sample of each herd/flock or was every animal on the holding inspected? 

Line 133 – change goat to ‘goats’

Line 144 – change regard to ‘regarding’ and were to ‘was’

Line 151 – resuscitate is the incorrect word here, ‘stimulate growth of’ may be a suitable alternative

Line 210 - change by a to ‘using a’

Line 211 – Line 211 insert the name of the software tool used for analysis

Line 251 – I feel the description of skin lesions needs some further clarification here.  Were all these lesions found to be abscess or when there some which had another form?  

Line 255 – C. pseudotuberculosis was isolated from all the sampled lesions.  Was this a pure growth in all cases, and if not were other organisms further identified to any extent or disregarded?  

Line 300 – insert …from ‘a’ sheep….

Line 320 - ‘resistance genes’ not resistant genes

Line 339 – …and ‘were’ clearly

Line 342 – possibly requires a reference to which published sequences 

Discussion Line 347 – 383 – as stated earlier one cannot justify conclusions around the wider prevalence of a disease in a country or region without first defining how your sample was selected.  If the animals with clinical external CLA were selected using a suitably randomised method it may be possible, but without this information broader conclusions are not valid.  

Line 388 – …which ‘has been’ attributed ‘by some authors’ to regional

Line 393 – …high ‘rates of’ resistance to….

Line 395 - …have reported ‘C. pseudotuberculosis isolates that were sensitive to the majority of antimicrobial classes’ and attributed….

Line 399 – change denoted to ‘noted’

Line 404 – delete were

Line 407 – …could pose a ‘potential threat’ to public health

Line 411 – change abscess to ‘abscesses’ 

Line 435 – I think known to be resistant to environmental stressors should be changed to possibly ‘has been shown to be highly persistent within the farm environment’

Line 444 – I am unsure what sharing of the same flock means.  Is it that sheep and goats are often grazed and housed together as one group? 

This study is based entirely on the identification of externally apparent lesions of CLA.  As no animals were euthanised for postmortem examination there are no data provided on animals which may have had internal lesions, but no external lesions.  This may have had an effect in reducing the apparent prevalence of disease within the sample population and would certainly be worth at least a mention in the discussion section.  

Generally the quality of the english within the paper is good, although some suggested edits have been included above.

Author Response

Reviewer #1

-----------------

R1.1: This paper describes detailed characterisation of Corynebacterium pseudotuberculosis isolates collected from cases of caseous lymphadentitis diagnosed in goats and sheep from smallholder herds/flocks in a region of Egypt.  A range of molecular techniques are employed.  In addition antibiotic sensitivity testing is also performed on these different isolates.  The results of this work are well presented and well structured in general. 

 AU: The authors thank the reviewer for considering our work well presented and structured. We have incorporated the following reviewer’s comments in the revised version of the manuscript.

R1.2: General concept comments: Article: In my opinion (and within the bounds of my experience) the description of the techniques employed is robust. The tables and figures are well constructed and easy to understand. I particularly liked the photographic images used here.

  AU: The authors thank the reviewer for his encouraging comment.

Specific comments/suggested edits:

R1.3: Line 6 – the numeral 5 has been added to the second name on this line

  AU: Thanks for noting that and 5 was deleted (Line 6).

R1.4: Line 28 – elsewhere the paper refers to a total of 220 animals being examined, not 200 as here. 

  AU: In total, 220 animals were included in the present study and 200 was unintentionally add in the abstract (Line 30).

R1.5: Lines 41-43 – I do not feel the final line of the summary is justified by what goes before. 

  AU: The authors agree with the reviewer comment and the conclusion line in the summary has been revised to reflect what goes before (Line 43-46).

R1.6: Line 47 – here and elsewhere ‘resistance genes’ not resistant genes

  AU: Done!  ... the resistant genes were changed throughout the manuscript.

R1.7: Line 68 – 72 – as above, in my opinion, without describing the method by which the smallholder herds/flocks were selected for examination, what proportion of the herd/flock was examined etc. it is impossible to draw any wider conclusions on disease prevalence.  Without this context, I am not persuaded that the findings, interesting as they may be, ‘emphasized the necessity of conducting monitoring and surveillance of CLA in the flocks of smallholder farmers.’

  AU: The reviewer made a valid point, and we concur with their comment regarding the unreliability of prevalence estimates without describing the method of smallholder flock selection. However, it is important to note that the estimation of CLA prevalence was not one of the study objectives. Our study focused on reporting the isolation rate of C. pseudotuberculosis from cases with enlarged lymph nodes. Thus, we made the required edits in the summary to make it clear for the readers (Line 43-46).

R1.8: Line 91 – C. pseudotuberculosis does not always infect humans should be changed.  Perhaps to ‘C. pseudotuberculosis may on rare occasions infect humans…’

  AU: The sentence was revised as recommended (Line 94-96).

R1.9: Line 107 – This reference is incorrect as the article cited makes no reference to virulence genes.  A check by the authors that all referencing within the paper is correct, may be warranted.

  AU: The authors would like to express gratitude to the reviewer for bringing this to their attention. Indeed, the referenced source is not relevant to the discussion of virulence genes. Consequently, a more appropriate reference has been utilized to replace it.

R1.10: Line 114 – change are to ‘is’  

  AU: Done! (Line 118).

  1. Materials & Methods

R1.11: Line 132 – This is described as a ‘cross-sectional study’ with no further detail provided. A description of how study herds/flocks were selected is required.  In addition as only 5-6 animals on average were examined at each site – was this a sub-sample of each herd/flock or was every animal on the holding inspected? 

  AU: Again, the reviewer made a very good point. In addition to the response to reviewer comment #R1.7, the authors would like to provide additional information about the study design and study population:

  • Smallholder flocks typically have a maximum of 10 animals.
  • All smallholders' flocks in the four districts were invited to participate, and out of those, only 39 smallholders agreed to join.
  • Clinical examinations were conducted on all animals in the 39 flocks, consisting of either sheep or goats. Animals with gross lesions, such as skin lesions (36/220) or enlarged lymph nodes (54/220), were reported.
  • It is important to note that CLA can manifest in different forms, with the external form being the most common. Therefore, clinical examination was considered a valuable tool for distinguishing between CLA-infected and non-infected cases.
  • Only animals with enlarged lymph nodes were sampled and subjected to bacterial culture and isolation.

Finally, we would like to confirm that the necessary edits have been made to clarify that the study did not calculate the prevalence of CLA in either sheep or goats (Lines 43-46; 73-75 and 144-149).

R1.12: Line 133 – change goat to ‘goats’

  AU: Done (Line 137).

R1.13: Line 144 – change regard to ‘regarding’ and were to ‘was’

  AU: Changed as suggested (Line 150).

R1.14: Line 151 – resuscitate is the incorrect word here, ‘stimulate growth of’ may be a suitable alternative

  AU: Changed as suggested (Line 158).

R1.15: Line 210 - change by a to ‘using a’

  AU: Changed as suggested (Line 216).

R1.16: Line 211 – Line 211 insert the name of the software tool used for analysis

  AU: Software name was added (Line 217).

R1.17: Line 251 – I feel the description of skin lesions needs some further clarification here.  Were all these lesions found to be abscess or when there some which had another form?  

  AU: Thirty-six of the total examined animals had different skin lesions regardless of the abscess, which are the main signs of caseous lymphadenitis. These skin lesions are due to shearing, castration, docking, umbilical, and tagging wounds. Sentence was updated to make it clear for the readers (Line 258-259).   

R1.18: Line 255 – C. pseudotuberculosis was isolated from all the sampled lesions.  Was this a pure growth in all cases, and if not were other organisms further identified to any extent or disregarded?  

  AU: Thank you for your insightful comment. As reported in the literature, C. pseudotuberculosis has been described as the primary pathogen of CLA in small ruminants worldwide. Several bacterial species are able to infect lymph nodes of small ruminants. Certain pathogens (Staphylococci, streptococci, and some enterobacteria species) are opportunistic in nature and may be found in soil, feces, water, and farm tools (e.g., enterobacteria); they are also commensal inhabitants of the skin and conjunctive and oral mucosae (e.g., staphylococci and streptococci) of livestock. However, we isolated C. pseudotuberculosis from closed abscesses in pure culture as a primary cause of CLA in sheep and goats, other infrequently bacterial agents were not detected

R1.19: Line 300 – insert …from ‘a’ sheep….

  AU: Done! (Line 310).

R1.20: Line 320 - ‘resistance genes’ not resistant genes

  AU: Changed throughout the manuscript as we indicated in reviewer comment #R1.6.

R1.21: Line 339 – …and ‘were’ clearly

  AU: Corrected (Line 350).

R1.22: Line 342 – possibly requires a reference to which published sequences 

  AU: The published sequences used in this study were sourced from the NCBI GenBank based on the nucleotide alignment of our sequences. The accession numbers for our sequences have been presented in the manuscript. This will facilitate readers' access to the published sequences, rendering the addition of sequence references unnecessary. The accession numbers provide comprehensive information required by the readers.

Discussion

R1.23: Line 347 – 383 – as stated earlier one cannot justify conclusions around the wider prevalence of a disease in a country or region without first defining how your sample was selected.  If the animals with clinical external CLA were selected using a suitably randomised method it may be possible, but without this information broader conclusions are not valid.  

  AU: Please see our response to reviewer comments #R1.7 and R1.11. The authors agree with the reviewer's comments. Although the estimation of CLA prevalence was not one of the study objectives, details regarding the selection of flocks and animals were provided in the study design and population section. Moreover, the authors deliberately refrained from drawing any broad conclusions regarding the disease prevalence (Lines 43-46; 73-75 and 144-149).

R1.24: Line 388 – …which ‘has been’ attributed ‘by some authors’ to regional

  AU: Done! (Line 406).

R1.25: Line 393 – …high ‘rates of’ resistance to….

  AU: Changed (Line 400).

R1.26: Line 395 - …have reported ‘C. pseudotuberculosis isolates that were sensitive to the majority of antimicrobial classes’ and attributed….

  AU: Changed as suggested (Line 417-418).

R1.27: Line 399 – change denoted to ‘noted’

  AU: Changed (Line 411).

R1.28: Line 404 – delete were

  AU: Deleted (Line 416).

R1.29: Line 407 – …could pose a ‘potential threat’ to public health

  AU: Done! (Line 429).

R1.30: Line 411 – change abscess to ‘abscesses’ 

  AU: Done! (Line 423).

R1.31: Line 435 – I think known to be resistant to environmental stressors should be changed to possibly ‘has been shown to be highly persistent within the farm environment’

  AU: Changed as suggested (Line 447).

R1.32: Line 444 – I am unsure what sharing of the same flock means.  Is it that sheep and goats are often grazed and housed together as one group? 

  AU: Smallholder flocks typically consist of up to 10 animals and are often raised in the backyard of their houses. Furthermore, smallholders commonly graze and house sheep and goats together.

R1.33: This study is based entirely on the identification of externally apparent lesions of CLA.  As no animals were euthanised for postmortem examination there are no data provided on animals which may have had internal lesions, but no external lesions.  This may have had an effect in reducing the apparent prevalence of disease within the sample population and would certainly be worth at least a mention in the discussion section.  

  AU: The authors would like to thank the reviewer for raising this point. Firstly, they acknowledge the validity of the reviewer's comment regarding the potential additional data that could be obtained through postmortem examinations, particularly in relation to the visceral form of CLA. However, it is crucial to acknowledge the challenge of convincing smallholders to euthanize their animals for such examinations. Secondly, it is important to note that the external or classical form of CLA is more prevalent, while the occurrence of the visceral form is less frequent. In light of the reviewer's feedback, the authors agree on the significance of highlighting this limitation in their study, and therefore, they have incorporated this point into the study's limitations section (Line 470-474).

Reviewer 2 Report

The article is very interesting especially since the subject of caseous lymphadenitis is currently very little studied. It is quite well written despite some linguistic errors. However there are some imperfections.

Author Response

Reviewer #2

-----------------

R2.1: The article is very interesting especially since the subject of caseous lymphadenitis is currently very little studied. It is quite well written despite some linguistic errors. However, there are some imperfections.

  AU: The authors thank the reviewer for considering our manuscript well written. We have incorporated the following reviewer’s comments in the revised version of the manuscript.

R2.2: Page 1 line 25 The present study aimed to isolate Corynebacterium and not C. pseudotuberculosis

  AU: Changed (Line 27).

R2.3: L28 24.54% (54/220) and not (54/200)

  AU: A total of 220 animals were included in the present study, and it was inadvertently mentioned as 200 in the abstract (Line 30).

R2.4: L30 92.6% were resistant to each of penicillin, and erythromycin. Please, review this sentence.

  AU: Sentence was rephrased to make it clear for the readers (Line 31-34).

R2.5: L 32 showed vancomycine resistance in place of « showed vancomycin resistant »

  AU: Changed (Line 35).

R2.6: L34 -lactam resistance gene was present in all isolates: only one beta-lactam resistance gene?

  AU: Sentence was rephrased to make it clear for the readers (Line 36-37).

R2.7: L41 biovar Ovis and not ovis

  AU: Corrected throughout the manuscript

R2.8: L83-84 as biotype Ovis (and not ovis) and 83 the second nitrate-positive strains classified as biotype Equi (and not equi). Please, correct these two transcriptions in all

the article.

  AU: Corrected throughout the manuscript

R2.9: L94-95 diagnosis of CLA; comma instead of semicolon

  AU: Done (Line 103).

R2.10: In the introduction, please talk about the serological methods (ELISA) widely used in some countries.

  AU: Reviewer raised a very good point and serological tests have been highlighted in the introduction (Line 99-100).

R2.11: L127 and resistant genes »

  AU: Changed throughout the manuscript.

R2.12: L144 gender in place of sex

  AU: Replaced (Line 150).

R2.13: L155 Gram-positive bacilli from small Gram-positive, white, dry, and

crumbly colonies »

  AU: Done (Line 162).

R2.14: L165 were incubated in place of « was incubated »

  AU: Done (Line 172).

R2.15: L167- of ethanol were added in place of « was added ».

  AU: Corrected (Line 174).

R2.16: p6 table 1: sul1 in place of Sul1

  AU: Changed.

R2.17: L175 tested included in place of « including »

  AU: Changed (Line 182).

R2.18: L197-198 anti-microbial resistance genes as erm(X) (macrolide resistance); erm(X) is a gene mediating resistance to macrolides, lincosamides and streptogramins

  AU: Thank you for bringing that to our attention. While our primary focus was on macrolide resistance, we acknowledge the reviewer's observation that the erm(X) gene not only mediates resistance to macrolides but also to lincosamides and streptogramins. We have made the necessary correction in the sentence to reflect this (Line 204).

R2.19: P8 table2 gender in place of sex

  AU: Corrected!

R2.20: L267 beta-subunit (rpoB) gene-specific PCR in place of b-sub-unit (rpoB) gene specific PCR.

  AU: Changed as suggested (Line 275).

R2.21: L273 and susceptibility to norfloxacin in place of and susceptible to norfloxacin

  AU: Changed (Line 281).

R2.22: Table 3 please add the signification of « MAR index »

  AU: Done! (Line 292).

R2.23: L285-288: use names of antibiotics in place of abbreviations

  AU: Abbreviations have been replaced as suggested (Line 294-298).

R2.24: Table 4 family names should be included: aminoglycosides - tetracyclines

sulfonamides

  AU: Done!

R2.25: L353 please mention the effect of shearing animals with contaminated instruments, dry climate, irritating vegetation, arthropods

  AU: Sentence revised as recommended (Line 365-366). 

R2.26: L401 that all of their C. pseudotuberculosis isolates were 100% sensitive to NOR and 20.3% were resistant to VAN : please write the name and not abbreviations of the antibiotics.

  AU: Done!.

R2.27: L403-404 with most of the isolates being resistant in place of « were resistant »

  AU: Changed.

R2.28: L246 65% of them were fed concentrates

  AU: Done.

R2.29: L263 On blood agar in place of « on blood ager », colonies were small, white, and

circular and beta hemolytic

  AU: Corrected.
